# The Longitudinal Association of Egg Consumption with Cognitive Function in Older Men and Women: The Rancho Bernardo Study

**DOI:** 10.3390/nu16010053

**Published:** 2023-12-23

**Authors:** Donna Kritz-Silverstein, Ricki Bettencourt

**Affiliations:** 1Herbert Wertheim School of Public Health and Longevity Science, University of California San Diego, La Jolla, CA 92093-0725, USA; 2Department of Family Medicine, School of Medicine, University of California San Diego, La Jolla, CA 92093-0725, USA; 3Division of Gastroenterology and Hepatology, School of Medicine, University of California San Diego, La Jolla, CA 92093-0725, USA; rbettencourt@health.ucsd.edu

**Keywords:** cognitive function, egg consumption, impaired cognitive function, longitudinal, memory, older men and women

## Abstract

This study examines the prospective association of egg consumption with multiple domains of cognitive function in older, community-dwelling men and women followed for 16.3 years. Participants were 617 men and 898 women from the Rancho Bernardo Cohort aged 60 and older, who were surveyed about egg intake/week in 1972–1974, and attended a 1988–1991 research visit, where cognitive function was assessed with 12 tests. Analyses showed that egg intake ranged from 0–24/week (means: men = 4.2 ± 3.2; women = 3.5 ± 2.7; *p* < 0.0001). In men, covariate-adjusted regressions showed that egg intake was associated with better performance on Buschke total (*p* = 0.04), long-term (*p* = 0.02), and short-term (*p* = 0.05) recall. No significant associations were observed in women (*p*’s > 0.05). Analyses showed that in those aged <60y in 1972–1974, egg intake was positively associated with scores on Heaton copying (*p* < *0*.04) and the Mini-Mental Status Exam (MMSE; *p* < *0*.02) in men and category fluency (*p* < 0.05) in women. Egg intake was not significantly associated with odds of poor performance on MMSE, Trails B, or category fluency in either sex. These reassuring findings suggest that there are no long-term detrimental effects of egg consumption on multiple cognitive function domains, and for men, there may be beneficial effects for verbal episodic memory. Egg consumption in middle age may also be related to better cognitive performance later in life.

## 1. Introduction

It has been estimated that in 2022, 6.5 million people in the US aged 65 years or older suffered from Alzheimer’s disease (AD), and an additional 13.1 million had mild cognitive impairment (MCI) [1,2]. Given the aging of the population, the prevalence of these disorders is expected to rise [1,2], making the identification of modifiable factors associated with the maintenance of cognitive function a public health priority.

Numerous studies examine lifestyle factors and generally report a decreased risk of AD and better cognitive function with higher educational attainment, light to moderate alcohol consumption, and higher levels of physical activity, cognitive engagement, and social support (see, for example, [3,4,5,6,7,8,9,10]). Other studies investigate dietary factors, including macro- and micronutrients, and generally report that low saturated fat consumption and high fruit and vegetable consumption were associated with decreased risk of AD and less cognitive decline (see reviews, [3,11,12]). Additionally, other studies report that choline and carotenoids, such as lutein and zeaxanthin, are associated with protective effects for cognitive function [13,14,15,16,17,18]. Eggs are low in saturated fat and have high levels of choline, carotenoids, and other micronutrients, but few studies directly investigate the association of egg consumption with cognitive function.

Previous cross-sectional studies reported either a protective effect or no association between egg intake and cognitive function. For example, higher egg consumption was associated with better cognitive function in a study of 317 Korean children aged 6–18 years [19] and in a study of 178 institutionalized men and women from Madrid aged 65 years and older [20]. Additionally, a study of 404 Chinese adults aged 60 years and older found that higher egg consumption was associated with decreased odds of MCI [21]. However, another study of 160 men and 204 women in China aged 90–105 (mean = 93) years found no significant difference in the frequency of egg consumption between those with and without MCI [22]. Likewise, a recent cross-sectional study using data from 2616 US adults aged 60 years and older enrolled in the National Health and Nutrition Survey (NHANES) found no association between egg consumption and a composite cognitive function score [23].

Only two previous studies examined the longitudinal associations of egg consumption with cognitive function but yielded conflicting results. One study of a subsample of 480 Finnish men aged 42–60 years when enrolled in the Kuopio Ischaemic Heart Disease Risk Factor Study found that higher intake of eggs at baseline was associated with better performance on the Trail Making Test (a measure of executive function), and a verbal fluency test when cognitive function was assessed 4 years later [24]. However, a study using a representative sample of 3835 US men and women aged 65 years and older followed over a 2-year period reported that egg consumption was not associated with measures of cognitive performance, including working memory, executive function, and global mental status [25]. Both studies had relatively short durations between the assessment of egg intake and the assessment of cognitive function and either did not include women or did not stratify analyses by sex. 

The purpose of this study is to examine the prospective association of egg consumption with multiple domains of cognitive function in a sample of 1515 older, community-dwelling men and women followed for an average of 16.3 years. It is hypothesized that higher egg consumption will be independently associated with better cognitive function in older men and women. It is also hypothesized that greater egg intake in middle age will be associated with better cognitive function at older ages. 

## 2. Materials and Methods

### 2.1. Participants 

Between 1972 and 1974, 6339 individuals, representing 82% of Rancho Bernardo, a predominantly Caucasian, middle-class, southern California community, were enrolled in a study of heart disease risk factors. These individuals have been followed ever since with almost yearly mailed questionnaires and periodic clinic visits. A total of 2212 individuals, representing 80% of the surviving community-dwelling participants, attended a follow-up clinic visit in 1988–1991 when cognitive function tests were first administered. After excluding those under 60 years of age (N = 365), those missing all cognitive function tests (N = 9), and those missing egg intake information from 1972 to 1974 (N = 323), there remained a total of 1515 individuals (617 men and 898 women) who formed the focus of this analysis.

This study was approved by the University of California San Diego (UCSD) Human Research Protections Program. All participants were ambulatory and gave written, informed consent prior to participation. 

### 2.2. Procedures

A self-administered questionnaire in 1972–1974 was used to obtain information on egg consumption. Specifically, participants were asked to write the answer to the question, “How many eggs do you eat per week? (visible eggs only)”. Information on demographic characteristics, including age, sex, education, and cigarette smoking history were obtained. Participants were queried about their history of heart attack (no/yes), stroke (no/yes), diabetes (no/yes), and high blood pressure (no/yes), along with whether they had taken any medication prescribed by a physician in the past week for high blood pressure (no/yes), high blood sugar (no/yes), and high cholesterol, triglycerides, or blood fats (no/yes). Systolic and diastolic blood pressures were measured in the participants’ left arm after they had been seated quietly for five minutes using a regularly calibrated standard mercury sphygmomanometer. Weight and height were measured, with participants wearing light clothing and no shoes, enabling the use of body mass index (BMI; weight(kg)/height(m)^2^) as an estimate of obesity. A blood sample was obtained by venipuncture after an overnight fast and sent to a CDC-certified laboratory for measurement of glucose, total cholesterol, and triglycerides. 

At the 1988–1991 research clinic visit, trained personnel administered 12 cognitive function tests individually to each participant. These tests were selected with help from the University of California San Diego (UCSD) Alzheimer’s Disease Research Center (ADRC) based on demonstrated reliability and validity [26,27]. Higher scores indicate better cognitive function except where noted. Cognitive function tests administered are summarized below as follows:

The Buschke-Fuld Selective Reminding Test [28] assesses verbal episodic memory and short- and long-term storage and retrieval of spoken words. Ten unrelated words were read to participants at a rate of one every 2 s. Immediately after, participants were asked to recall the entire list, reminded of any words they missed, and asked to recall the entire list again. This procedure was followed for six trials. Points were based on the number of items and trials needed for recall. Words were determined to be in long-term storage starting on the first trial on which the word was consecutively recalled or in short-term storage, as well as the overall total number of words recalled across all trials. Thus, measures of long- and short-term memory and total recall were obtained. Higher scores on short-term memory indicate poorer performance because they reflect words not successfully encoded into long-term storage.

The Heaton Visual Reproduction Test [29], adapted from the Wechsler Memory Scale [30], assesses memory for geometric forms. Three cards (stimuli) of increasingly complex geometric figures were presented to participants one at a time, for 10 s each. Participants were asked to reproduce the figures on the card immediately to assess short-term memory and after a 30 min delay of unrelated testing to assess long-term memory for geometric forms. Afterward, participants were asked to copy each stimulus figure, which enabled the assessment of visuospatial impairments. Thus, three scores were obtained: immediate recall; delayed recall; and copying.

The Mini-Mental State Examination (MMSE) [31,32], a test of global function, assesses orientation, registration, attention, calculation, language, and recall. Total MMSE scores can range from 0 to 30; persons with dementia usually score ≤ 24. Two MMSE subtests were analyzed separately: counting backward from 100 by sevens (Serial 7′s), which assesses calculation and concentration; and spelling the word “world” backward (world backward), which assesses attention, both of which are indicative of working memory. The maximum score was 5 for each. 

Two items from the Blessed Information-Memory-Concentration Test [27] assessed executive function and concentration by having the participant name the months of the year backward and assessed verbal episodic memory by asking participants to recall a five-part name and address following a 10-minute delay. Participants were given two points for correctly naming the months of the year backward on the first attempt and one point if successful on a second attempt; otherwise, no points were given. Participants were given one point for each part of the name/address recalled correctly. The maximum score was 7. 

The Trail-making Test, part B (Trails B), from the Halstead-Reitan Neuropsychological Test Battery [33], is a test of executive function that assesses visuomotor tracking and mental flexibility. Participants scanned a page containing letters and numbers within circles and were asked to connect numbers and letters in ascending order, alternating between numbers and letters (e.g., 1 to A to 2 to B to 3 to C, and so on). A maximum of 300 s was allowed; performance was rated by time (in seconds) required to finish the test; higher scores indicated poorer performance.

Category Fluency [34] is a test of verbal fluency that assesses semantic memory and executive function. Participants were asked to name as many animals as possible in 1 min. The score is the number of animals correctly named; repetitions, variants (e.g., dogs after producing dog), and intrusions (e.g., apple) were not counted. 

### 2.3. Statistical Analysis

Education was dichotomized as high school or lower vs. some college or higher. Cigarette smoking status was dichotomized into current smoking (no/yes). Because of skewness, triglycerides were presented as medians, with values logged for testing purposes. Means and distributions for continuous variables and rates for categorical variables were calculated. Comparisons of variables by sex were performed with independent *t*-tests for continuous variables and chi-square analyses for categorical variables. Because of the significant sex differences found, all further analyses were sex-specific. Sex-specific comparisons of characteristics by categorical egg consumption were performed with analysis of variance for continuous variables and chi-square analysis or Fisher’s exact test for categorical variables. Potential confounders of the egg consumption—cognitive function association—were identified based on being known confounders in the literature and based on their associations with both egg intake and cognitive function in this study. Linear regression analysis was used to examine the associations of egg consumption as a continuous variable with cognitive function after adjustment for potentially confounding covariates. Analyses were repeated after restriction to those aged <60 years at enrollment to determine the long-term effect of egg intake in middle age on cognitive function.

Cognitive function scores were also analyzed as categorical outcomes using cutoffs indicative of poor performance recommended by the UCSD ADRC. Cutoffs were available for five tests: Buschke Selective Reminding Test long-term memory (≤13); Heaton Visual Reproduction Test immediate recall (≤7); MMSE (≤24); Trails B (≥132); and Category Fluency (≤12). However, because <1.5% scored below the Buschke long-term recall cutoff and ≤3.2% scored below the Heaton Visual Reproduction Test cutoff, the results are presented only for the MMSE, Trails B, and Category Fluency. Sex-specific logistic regression analysis was used to examine the association of egg consumption with odds of poor cognitive function on the MMSE, Trails B, and category fluency after adjustment for potentially confounding covariates. 

In sensitivity analyses, comparisons of baseline characteristics between those who came to both visits and those who did not attend the 1988–1991 clinic visit were performed with independent *t*-tests, Wilcoxon, and chi-square analysis to examine survival bias. 

Statistical analyses were performed with SAS (version 9.4, SAS Institute, Cary, NC, USA); all tests were two-tailed, with *p*-value ≤ 0.05 considered significant.

## 3. Results

In this sample, the average length of follow-up was 16.3 ± 0.8 years (range = 13.9–19.1 years). Comparisons of characteristics from 1972 to 1974 (Table 1) showed that men had significantly higher body mass index (*p* < 0.0001), glucose (*p* < 0.0001), triglycerides (*p* < 0.0020), education (*p* < 0.0001), and rates of self-reported diabetes (*p* = 0.0072), but lower total cholesterol (*p* < 0.0001) and rates of current smoking (*p* < 0.0060) than women. There were no significant differences between men and women in the ages of 1972–1974 (means = 59.2 and 59.0, respectively; *p* = 0.5626). Significant sex differences were observed in almost all cognitive function tests (Table 1). Men performed significantly better than women on the Heaton immediate and delayed recall tasks, serial 7’s, Trails B, and category fluency, but worse on Buschke total recall, long-term recall, short-term recall, the MMSE, world backward, and the Blessed items. 

In both men and women, egg intake ranged from 0 to 24/week. However, men consumed significantly more eggs per week than women (means = 4.2 ± 3.2 vs. 3.5 ± 2.7, respectively; *p* < 0.0001). Rates of egg consumption per week varied by sex; greater proportions of men consumed eggs at the higher levels, whereas greater proportions of women consumed eggs at the lower levels (see Figure 1). For instance, in men, 5.5% consumed no eggs/week, and 18.0% consumed seven or more eggs/week, whereas in women, 9.9% consumed no eggs/week, and 13.0% consumed seven or more eggs per week. Because of the significant sex differences in demographic characteristics, egg consumption per week, and cognitive function, all further analyses performed were sex-specific. 

For both sexes, unadjusted comparisons of characteristics by egg consumption (Table 2) showed that those who consumed seven or more eggs/week had lower mean cholesterol and triglycerides, although differences overall were not statistically significant (*p*’s > 0.05). Rates of cholesterol-lowering medication use were the lowest among those consuming seven or more eggs/week but were too low overall for valid statistical comparisons. Mean glucose levels were lowest for those consuming four or five–six eggs/week in both men (*p* = 0.0465) and women (*p* = 0.0643). Other differences in characteristics by categorical egg consumption are shown in Table 2.

Unadjusted comparisons of cognitive function test scores by categorical egg consumption (Table 3) showed that among men, those with higher levels of egg intake performed better on Buschke total (*p* = 0.0527), long-term (*p* = 0.0316) and short-term recall (*p* = 0.0391), and the MMSE (*p* = 0.0018), but worse on Trails B (*p* = 0.0194). Men who consumed two or three eggs per week performed worse on the Heaton copying task and on the Blessed items than those with higher or lower egg intake (*p*’s < 0.05). Among women, unadjusted comparisons showed that those with higher levels of egg intake performed better on Buschke total recall (*p* = 0.0115), long-term recall (0.0206), and Heaton immediate (*p* = 0.0220) and delayed (*p* = 0.0021) visual recall tasks. Women who consumed seven or more eggs/week also performed better on Trails B and category fluency than those who did not consume eggs (*p*’s < 0.05). 

Among men, age and education-adjusted regression analyses (Table 4) showed that egg intake as a continuous variable was significantly associated with better performance on Buschke total, long-term, and short-term recall. These associations remained significant after additional adjustment for smoking, cholesterol level, use of cholesterol-lowering medications, and history of heart attack and hypertension (B = 0.22, *p* = 0.0415 for Buschke total recall, B = 0.33, *p* = 0.0245 for long-term recall, and B = −0.12, *p* = 0.0494 for short-term recall). Thus, each additional egg consumed per week was associated with an increase of 0.22 in total recall score, 0.33 in long-term recall score, and −0.12 (indicating better performance) in short-term recall score. No other associations were observed between egg intake and cognitive function in men. Among women, no significant associations were observed between egg intake and scores on any of the cognitive function tests and adjustment for age, education, and other covariates (*p*’s > 0.05).

Restricting analyses to those aged younger than 60 years at enrollment (Table 5) showed that greater egg consumption as assessed in middle age was associated with somewhat better performance on most cognitive function tests approximately 16 years later, with significantly higher scores on the Heaton copying subtest (B = 0.061, *p* = 0.0352) and the MMSE (B = 0.058, *p* = 0.0154), but worse scores on Trails B (B = 1.749, *p* = 0.0110) for men. Among women, greater egg consumption in middle age was associated with significantly higher scores on category fluency (B = 0.145, *p* = 0.0464). 

Results of logistic regression analyses examining the adjusted odds of categorical poor performance on cognitive function tests by egg consumption are shown in Table 6. For both men and women, egg intake per week was not significantly associated with increased odds of poor performance on the MMSE, Trails B, or category fluency.

Sensitivity analyses examining the possibility of survivor bias showed that compared to participants who attended both enrollment and follow-up visits, those who only attended the enrollment visit were significantly older (*p* < 0.001), had higher total cholesterol, triglycerides (*p*’s < 0.001), and higher rates of cholesterol-lowering medication use (*p* < 0.001). However, there were no differences in education or rates of current smoking (*p*’s > 0.10).

## 4. Discussion

### 4.1. Study Outcomes

In this cohort of community-dwelling individuals with cognitive function assessed more than 16 years after the assessment of egg intake, analyses showed that for men, greater egg consumption was associated with small but significantly better performance on total recall, short-term, and long-term memory. Specifically, each egg consumed per week was associated with increases of 0.22 in total memory and 0.33 in long-term memory and decreases of 0.12 (indicating better performance) in short-term memory. These associations were independent of age, education, cigarette smoking, cholesterol level, use of cholesterol-lowering medications, and histories of heart attacks and hypertension, and adjusting for these covariates did not alter the associations. No other differences in cognitive function were observed for men, and no association of egg consumption with cognitive function was observed in women. Although differences were small, analyses restricted to those younger than 60 at enrollment suggested that egg consumption in middle age was associated with better performance on some cognitive function tests later in life. Egg consumption was not related to the odds of impaired cognitive function in either men or women.

The results of this study are in accord with those from a subsample of 480 Finnish men from the Kuopio Ischaemic Heart Disease Risk Factor Study, which found significant prospective associations of egg intake with cognitive function [24]. However, that study reported significant positive associations between egg intake and scores on Trails A and verbal fluency after 4 years of follow-up [24], whereas the current study found positive associations with total, short-term, and long-term recall in men after a longer, 16-year follow-up. 

Results of this study disagree with those of Bishop and Zuniga, who, using a nationally representative sample of 3835 men and women aged 65 and older from the Health and Retirement Study and the Health Care and Nutrition Study, found that egg intake assessed in 2013 was not associated with any measure of cognitive function over a two-year period [25]. However, in that study, cognitive function was assessed via a telephone interview, whereas the current study involved in-person, face-to-face assessments. Additionally, unlike the present study, that study did not stratify analyses by sex, possibly obscuring differences in associations for men and women. Guidelines to limit dietary cholesterol were first suggested by the American Heart Association in 1968 and then widely adopted by other agencies by 1995 [35]. Thus, unlike the present study, which collected egg intake data from 1972 to 1974 at a time temporally close to the initiation of these guidelines and potentially prior to their widespread adoption, Bishop and Zuniga assessed egg intake at a time when guidelines limiting egg and cholesterol intake were widely known and had existed for over 40 years. Egg intake in the Bishop and Zuniga study [25] was much lower than in the present study—only 0.34/day on average or 2.4/week compared to 3.5/week in women and 4.2/week in men observed here. This, coupled with the relatively short follow-up, may have made it difficult to detect associations.

To our knowledge, this is the first prospective study to examine associations of egg intake with cognitive function separately for men and women. Although some associations were observed in men after adjustment for covariates, no associations were observed between egg intake and any of the cognitive function measures in women. It is possible that this lack of observed associations in women may be due to the fact that, despite having a similar range in the number of eggs consumed per week (0–24), greater proportions of men than women consumed eggs at the higher levels, and the smaller variability of egg consumption in women may have led to attenuated, non-significant associations.

In this study, we found evidence that egg consumption in middle age might be associated with somewhat better cognitive function later in life. This is a novel finding that warrants further study with larger samples of middle-aged and younger adults as it suggests a potential long-term impact of egg consumption on cognitive health. 

This study, which had the longest duration between assessments of egg consumption and cognitive function, also found no associations between egg intake and categorically defined impaired cognitive function in either men or women. These results are in accord with a cross-sectional study of 870 community-dwelling Chinese men and women aged 90 and older, which found no association between egg intake and mild cognitive impairment based on MMSE score [22]. However, our results are in contrast to those of a cross-sectional study of 404 men and women aged 60 and older in Beijing, which reported that a higher daily intake of eggs was associated with lower odds of mild cognitive impairment [21].

### 4.2. Biologic Plausibility

It is biologically plausible that egg consumption is associated with beneficial effects on cognitive function. Although eggs are a lipid-rich food that is high in dietary cholesterol, they are low in saturated fat, which may have neuroprotective effects on the brain [36]. Additionally, eggs are an excellent source of protein and amino acids, as well as nutrients and bioactive compounds [37]. Eggs also contain high levels of choline, a nutrient needed to produce acetylcholine, a neurotransmitter important for memory [37]. A study of 1991 adults aged 36–83 years from the Framingham Offspring Study, with higher choline intake, had better verbal and visual memory [38]. Similarly, a more recent study among 1258 Finnish men from the Kuopio Ischaemic Heart Disease Risk Factor Study reported that higher choline intake was associated with better performance on tests of verbal fluency, verbal memory, and visual memory [14]. Eggs are also a good source of lutein and zeaxanthin, carotenoids that are important for cognitive function in the elderly [18,39]. A study of 78 octogenarians and 220 centenarians from the Georgia Centenarian Study showed that higher serum and brain concentrations of lutein and zeaxanthin were associated with better performance on measures of memory, executive function, and language [18]. The protective effect of these carotenoids for cognitive function may be due to anti-oxidant or anti-inflammatory actions in the brain [18]. Small clinical trials of 12–15 institutionalized individuals with AD reported that those given lutein and zeaxanthin had less progression of AD [40]. Additionally, a small clinical trial of 51 older community-dwelling adults reported that those given 12 mg of lutein and zeaxanthin had improved attention, cognitive flexibility, and performance in other tests [41].

### 4.3. Limitations and Strengths

Several limitations of this study were considered. Rancho Bernardo Study participants are predominantly Caucasian, well-educated, and have good access to medical care, which may limit generalizability. Given the number of statistical tests performed, the possibility that observed differences were due to chance cannot be excluded. However, all comparisons were a priori attempts to understand the contradictory literature on egg consumption and cognitive function, and the sex-specific associations with egg consumption were consistently observed for the same tests. We also cannot exclude participation bias, as in any sample of older adults where those with the most impaired cognitive function do not participate. We also cannot exclude survival bias, where the oldest and least healthy may have died prior to participation. However, these biases would yield conservative estimates of any true association. Finally, no brain imaging studies were performed at this visit, so we are unable to relate differences in cognitive function with egg consumption to underlying changes in the brain.

This study has several strengths, however, including the large sample size of community-dwelling older adults and the use of a battery of standardized measures, which allowed for the measurement of multiple domains of cognitive function. The long follow-up and focus on those aged 60 years and older at cognitive assessment ensured the variation in cognitive function. Additionally, baseline assessment of egg intake was relatively close temporally to the introduction of the AHA guidelines suggesting limitation of egg and dietary cholesterol intake, and prior to the widespread use of statins, thus limiting potential bias from these factors. Finally, the homogeneity of this cohort is also an advantage as there is less confounding of test performance due to socio-cultural differences, education, or access to health care.

## 5. Conclusions

The results of this study are reassuring and suggest that there are no long-term detrimental effects of egg consumption on cognitive impairment or on multiple domains of cognitive function, and there may be beneficial effects for verbal episodic memory for men. Results also suggest that greater egg intake in middle age is associated with better cognitive performance at older ages. Eggs represent a readily available, relatively inexpensive source of protein and other nutrients that are beneficial for cognitive health as well as overall health. Future studies with large samples of men and women are needed to confirm the sex-specific associations observed here and to examine associations in large samples of younger and middle-aged individuals followed across the lifespan. Additionally, brain imaging studies are needed to determine if differences in cognitive function with egg consumption are related to underlying brain changes.

## Figures and Tables

**Figure 1 nutrients-16-00053-f001:**
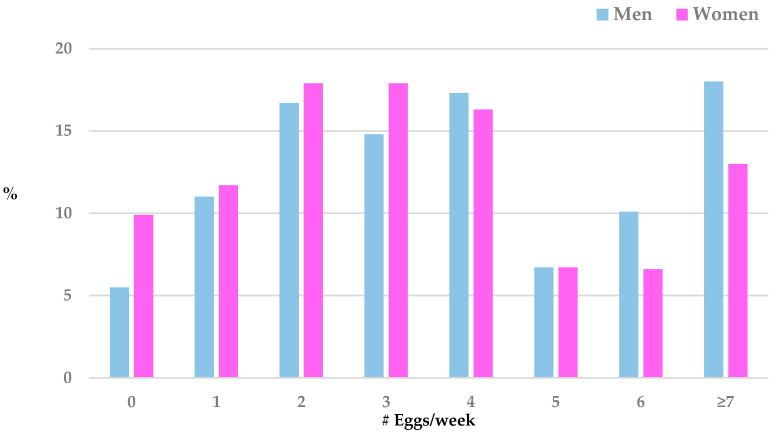
Rates of egg consumption/week for men and women; Rancho Bernardo Cohort, 1972–1974.

**Table 1 nutrients-16-00053-t001:** Characteristics in 1972–1974 and cognitive function in 1988–1991 for men and women; Rancho Bernardo, CA, USA.

	Overall	Men (N = 617)		Women (N = 898)		
**Characteristic**	**N**	**N**	**(%)**	**N**	**(%)**	***p*-Value ^a^**
Education (some college)	1502	476	(78.3)	564	(63.1)	**<0.0001**
Current Smoking	1515	115	(18.6)	221	(24.6)	**0.0060**
Cholesterol-lowering Meds	1515	18	(2.9)	25	(2.8)	0.8779
Diabetes (self-report)	1510	23	(3.7)	14	(1.6)	**0.0072**
Heart attack (self-report)	1505	27	(4.4	17	(1.9)	**0.0085**
Stroke (self-report)	1505	5	(0.008)	2	(0.002)	0.2088
Hypertension (self-report)	1505	119	(19.3)	181	(20.4)	0.6468
		**Mean**	**(SD)**	**Mean**	**(SD)**	***p*-Value ^a^**
Age (years)	1515	59.2	(8.3)	59.0	(7.8)	0.5626
Body Mass Index (BMI)	1515	25.8	(2.8)	23.4	(3.1)	**<0.0001**
Total Cholesterol	1512	210.5	(33.5)	221.1	(37.6)	**<0.0001**
Glucose	1448	107.9	(19.4)	103.1	(18.1)	**<0.0001**
Triglycerides ^b^	1512	109.0	(78.0)	102.1	(60.0)	**0.0020**
Systolic Blood Pressure	1515	133.0	(24.0)	128.1	(24.6)	**<0.0001**
Diastolic Blood Pressure	1515	79.2	(12.0)	77.2	(11.7)	**<0.0001**
# eggs/week	1515	4.2	(3.2)	3.5	(2.7)	**<0.0001**
Cognitive Function Tests						
Buschke	1464					
Total Recall		34.3	(9.7)	39.2	(9.0)	**<0.0001**
Long-term Recall		26.4	(12.8)	33.2	(12.3)	**<0.0001**
Short-term Recall ^c^		7.9	(4.6)	6.1	(4.2)	**<0.0001**
Heaton	1477					
Immediate Recall		9.8	(3.8)	9.1	(3.4)	**0.0002**
Delayed Recall		7.3	(4.5)	6.4	(3.9)	**<0.0001**
Copying		15.1	(2.2)	15.1	(2.0)	0.9801
Mini-Mental State Exam	1492	26.8	(2.5)	27.3	(1.9)	**<0.0001**
Serial 7’s	1476	4.3	(1.1)	3.9	(1.3)	**<0.0001**
“World” Backward	1500	4.7	(0.9)	4.8	(0.6)	**<0.0001**
Blessed Items	1501	5.9	(1.6)	6.1	(1.3)	**0.0428**
Trails B ^c^	1473	132.2	(64.5)	144.2	(67.5)	**0.0007**
Category Fluency	1504	18.3	(5.4)	17.1	(4.8)	**<0.0001**

^a^ *p*-values from independent *t*-tests or chi-square analysis; no adjustments for multiple comparisons. ^b^ Triglycerides shown as median (IQR). Original values were logged for testing purposes. ^c^ For Buschke short-term recall and Trails B, lower scores indicate better performance. Bolded p-values indicate statistical significance.

**Table 2 nutrients-16-00053-t002:** Unadjusted comparisons of characteristics by categorical egg consumption in 1972–1974.

	Egg Consumption 1972–1974							
	**0/Week**	**1/Week**	**2/Week**	**3/Week**	**4/Week**	**5–6/Week**	**≥7/Week**	** *p* ** **-Value ^a^**
**Men**	**N = 34**	**N = 68**	**N = 103**	**N = 91**	**N = 107**	**N = 103**	**N = 111**	
Age, mean (sd)	60.8 (8.4)	60.2 (8.7)	60.8 (8.4)	60.2 (8.7)	57.9 (7.9)	59.0 (8.8)	59.1 (8.0)	0.1315
BMI, mean (sd)	26.1 (2.6)	25.5 (2.8)	25.8 (2.9)	25.5 (2.3)	25.8 (2.7)	25.9 (2.8)	25.8 (3.3)	0.8877
Cholesterol, mean (sd)	218.1 (33.9)	213.6 (35.2)	214.6 (32.7)	212.5 (36.2)	206.2 (33.1)	212.5 (34.7)	203.0 (28.6)	0.0668
Glucose, mean (sd)	111.9 (27.4)	113.6 (21.6)	109.0 (21.9)	106.3 (16.8)	104.5 (15.7)	105.6 (15.7)	109.1 (21.0)	**0.0465**
Triglycerides ^b^, mdn (IQR)	94.0 (60.0)	119.5 (75.0)	112.5 (88.0)	123.0 (73.5)	114.0 (75.0)	107.0 (72.0)	96.0 (74.0)	0.3940
College, n (%)	24 (75.0)	45 (67.2)	81 (81.0)	76 (84.4)	82 (78.1)	75 (72.8)	93 (83.8)	0.0792
Smoking, n (%)	3 (8.8)	14 (20.6)	11 (10.7)	18.(19.8)	25 (23.4)	19 (18.4)	25 (22.5)	0.1526
Chol-lowering meds n (%)	5 (14.7)	3 (4.4)	5 (4.8)	2 (2.2)	1 (0.9)	1 (1.0)	1 (0.9)	--- ^c^
Diabetes, n (%)	1 (2.9)	2 (2.9)	3 (2.9)	3 (3.3)	4 (3.7)	3 (2.9)	7 (6.4)	--- ^c^
Hx heart attack, n (%)	2 (5.9)	4 (5.9)	8 (7.8)	1 (1.1)	2 (1.9)	7 (6/8)	3 (2.7)	--- ^c^
Hx hypertension n (%)	7 (20.6)	9 (14.3)	16 (15.5)	27 (30.0)	10 (17.8)	23 (22.3)	18 (16.4)	0.1294
	**0/Week**	**1/Week**	**2/Week**	**3/Week**	**4/Week**	**5–6/Week**	**≥7/Week**	** *p* ** **-Value ^a^**
**Women**	**N = 89**	**N = 105**	**N = 161**	**N = 161**	**N = 146**	**N = 119**	**N = 117**	
Age, mean (sd)	59.3 (7.5)	60.8 (7.9)	60.0 (7.5)	59.3 (7.8)	58.5 (7.7)	57.7 (8.3)	57.1 (7.8)	**0.0036**
BMI, mean (sd)	23.5 (3.6)	23.4 (3.3)	23.5 (2.9)	23.3 (3.2)	23.7 (3.1)	23.7 (3.1)	23/0 (2.7)	0.8877
Cholesterol, mean (sd)	225.4 (42.7)	226.1 (39.3)	220.5 (34.1)	223.5 (38.1)	220.2 (35.4)	218.5 (39.8)	214.6 (35.3)	0.2457
Glucose, mean (sd)	102.6 (13.8)	105.6 (15.6)	100.9 (15.4)	103.0 (14.3)	102.5 (17.7)	101.0 (15.3)	107.2(29.8)	0.0643
Triglycerides ^b^, mdn (IQR)	113.0 (68.0)	107.0 (55.0)	103.0 (62.0)	101.0 (67.0)	99.0 (52.0)	96.0 (58.0)	95.0 (57.0)	0.1198
College, n (%)	50 (56.8)	64 (60.9)	93 (57.8)	107 (66.9)	87 (59.6)	79 (66.4)	84 (73.0)	0.0930
Smoking, n (%)	21 (23.6)	21 (20.0)	35 (21.7)	22 (20.5)	35 (24.0)	31 (26.0)	45 (38.5)	**0.0155**
Chol-lowering meds, n (%)	10 (11.2)	4 (3.8)	3 (1.9)	1 (0.6)	2 (1.4)	4 (3.3)	1 (0.9)	--- ^c^
Diabetes, n (%)	2 (2.2)	2 (1.9)	1 (0.6)	2 (1.2)	4 (2.8)	2 (1.7)	1 (0.9)	--- ^c^
Hx heart attack, n (%)	4 (4.5)	4 (3.8)	1 (0.6)	1 (0.6)	4 (2.8)	2 (1.7)	1 (0.9)	--- ^c^
Hx hypertension, n (%)	22 (25.0)	24 (22.9)	25 (15.6)	38 (23.7)	30 (20.5)	29 (24.6)	23 (19.7)	0.4819

BMI = body mass index; Chol = Cholesterol; Hx = history ^a^ *p*-values from analysis of variance, chi-square, or Fisher’s Exact Test. No adjustments for multiple comparisons. ^b^ Triglycerides shown as median (mdn) and interquartile range (IQR). Original values were logged for testing purposes. ^c^ Rates too low for valid statistical comparisons. Bolded *p*-values indicate statistically significant.

**Table 3 nutrients-16-00053-t003:** Unadjusted comparisons of cognitive function scores by egg consumption in 1972–1974 in men and women.

Egg Consumption 1972–1974								
	**0/Week**	**1/Week**	**2/Week**	**3/Week**	**4/Week**	**5–6/Week**	**≥7/Week**	
**Men**	**N = 34**	**N = 68**	**N = 103**	**N = 91**	**N = 107**	**N = 103**	**N = 111**	***p*-Value ^a^**
Buschke								
Total Recall	33.2	33.8	31.8	33.6	35.9	34.8	35.5	**0.0527**
Long-Term Recall	25.6	26.0	22.5	26.0	28.0	28.5	28.5	**0.0316**
Short-Term Recall ^b^	7.6	7.8	9.2	7.5	7.9	7.0	7.0	**0.0391**
Heaton								
Immediate Recall	10.1	9.7	9.2	9.2	10.5	10.3	10.3	0.1471
Delayed Recall	7.5	7.1	6.6	6.4	8.3	7.5	7.5	0.0720
Copying	15.8	14.9 *	14.7 *	14.6 *	15.4	15.3	15.3	**0.0311**
MMSE	26.7	26.8	26.3	26.0	27.4	26.9	26.9	**0.0018**
Serial 7’s	4.3	4.2	4.3	4.1	4.5	4.4	4.4	0.0763
“World” Backward	4.8	4.9	4.5	4.5	4.8	4.6	4.6	0.0577
Blessed Items	6.3	6.1	5.6 *	5.3 *	6.1	6.0	6.0	**0.0004**
Trails B ^b^	124.1	119.3	147.7	136.5	118.3	138.6	138.6	**0.0194**
Category Fluency	18.9	18.5	17.1	17.7	18.9	18.4	18.4	0.1843
	**0/Week**	**1/Week**	**2/Week**	**3/Week**	**4/Week**	**5–6/Week**	**≥7/Week**	
**Women**	**N = 89**	**N = 105**	**N = 161**	**N = 161**	**N = 146**	**N = 119**	**N = 117**	***p*-Value ^a^**
Buschke								
Total Recall	37.8	38.0	39.3	39.2	38.1	40.1	41.7 *	**0.0115**
Long-Term Recall	31.3	31.5	33.7	33.1	31.7	33.9	36.5 *	**0.0206**
Short-Term Recall ^b^	6.5	6.7	5.7	6.1	6.5	6.2	5.3 *	0.1136
Heaton								
Immediate Recall	9.0	8.7	8.8	8.8	9.3	9.1	10.2 *	**0.0220**
Delayed Recall	7.0	5.3 *	6.2	5.9 *	6.5	6.7	7.3	**0.0021**
Copying	15.4	14.8	14.9	15.0	15.1	15.0	15.3	0.2615
MMSE	27.1	27.1	27.3	27.2	27.2	27.6	17.4	0.3649
Serial 7’s	3.7	4.0	4.0	3.8	3.8	4.0	4.2 *	0.0913
“World” Backward	4.9	4.8	4.9	4.8	4.8	4.9	4.9	0.9195
Blessed Items	5.9	6.1	6.1	6.0	6.0	6.2	6.1	0.7219
Trails B ^b^	155.6	155.5	138.9	148.4	138.0	145.2	133.1 *	0.0784
Category Fluency	16.6	16.7	17.1	16.7	17.0	17.7	18.3 *	0.1587

^a^ Overall *p*-values from analysis of variance: No adjustments for multiple comparisons; ^b^ Higher scores indicate poorer performance. * Indicates individual mean statistically different (*p* < 0.05) from referent category (0/week). Bolded p-values indicate statistically significant.

**Table 4 nutrients-16-00053-t004:** Sex-specific adjusted associations of 1972–1974 egg intake with 1988–1991 cognitive function.

			Men		Women	
			B	*p*-Value	B	*p*-Value
Buschke	Total recall	Model 1	0.207	**0.0475**	0.133	0.2008
		Model 2	0.209	**0.0447**	0.130	0.2133
		Model 3	0.216	**0.0415**	0.114	0.2785
	Long-term recall	Model 1	0.318	**0.0245**	0.141	0.3223
		Model 2	0.323	**0.0224**	0.139	0.3316
		Model 3	0.333	**0.0204**	0.113	0.4300
	Short-term recall ^a^	Model 1	−0.109	0.0578	−0.018	0.7306
		Model 2	−0.111	0.0534	−0.018	0.7731
		Model 3	−0.115	**0.0494**	−0.007	0.8864
Heaton	Immediate recall	Model 1	0.066	0.1173	0.060	0.1359
		Model 2	0.068	0.1061	0.061	0.1315
		Model 3	0.065	0.1234	0.056	0.1677
	Delayed recall	Model 1	0.085	0.0890	0.048	0.2950
		Model 2	0.087	0.0810	0.048	0.2887
		Model 3	0.073	0.1471	0.046	0.3153
	Copying	Model 1	0.027	0.3141	−0.013	0.6011
		Model 2	0.029	0.2808	−0.010	0.6880
		Model 3	0.034	0.2067	−0.013	0.6024
Mini-Mental State Exam		Model 1	0.043	0.1344	0.006	0.7719
		Model 2	0.044	0.1231	0.007	0.7444
		Model 3	0.050	0.0874	0.004	0.8418
Serial 7’s		Model 1	0.017	0.2026	0.029	0.0816
		Model 2	0.017	0.2094	0.031	0.0598
		Model 3	0.018	0.1946	0.028	0.0912
“World” Backward		Model 1	−0.003	0.7644	−0.003	0.6652
		Model 2	−0.003	0.7740	−0.003	0.6524
		Model 3	−0.002	0.8254	−0.002	0.7639
Blessed Items		Model 1	0.020	0.2817	−0.004	0.8061
		Model 2	0.020	0.2895	−0.004	0.8146
		Model 3	0.025	0.1978	−0.005	0.7431
Trails B ^a^		Model 1	1.418	0.0505	0.002	0.9976
		Model 2	1.389	0.0554	−0.078	0.9193
		Model 3	1.187	0.1061	0.219	0.7771
Category Fluency		Model 1	0.033	0.5865	0.096	0.0819
		Model 2	0.034	0.5798	0.101	0.0690
		Model 3	0.050	0.4193	0.100	0.0743

Multiple linear regression results. B = sex-specific beta coefficients. ^a^ Higher scores indicate poorer performance. Model 1—adjusted for age at 1972-74, college education; Model 2—adjusted for Model 1 variables plus current smoking; Model 3—adjusted for Model 2 variables plus total cholesterol, self-reported use of cholesterol-lowering medication, history of heart attack, and history of hypertension. Bolded p-values indicate statistically significant.

**Table 5 nutrients-16-00053-t005:** Sex-specific adjusted * associations of egg consumption among those aged <60 years (middle-aged) in 1972–1974 with cognitive function test scores in 1988–1991.

	Men		Women	
	B	*p*-Value	B	*p*-Value
Buschke				
Total recall	0.226	0.0866	0.041	0.7419
Long-term recall	0.284	0.1246	0.022	0.8997
Short-term recall ^a^	−0.064	0.3729	0.001	0.9915
Heaton				
Immediate recall	0.073	0.1905	0.077	0.1346
Delayed recall	0.115	0.0883	0.075	0.2009
Copying	0.061	**0.0352**	−0.036	0.1989
Mini-Mental State Exam	0.058	**0.0154**	−0.0002	0.9266
Serial 7’s	0.008	0.6067	0.033	0.1138
“World” backward	0.003	0.7930	−0.006	0.4947
Blessed items	0.025	0.1463	−0.012	0.5040
Trails B ^a^	1.749	**0.0110**	0.311	0.7196
Category Fluency	0.030	0.7021	0.145	**0.0464**

* Results of regression analyses adjusted for age in 1972–1974, college education, current smoking, total cholesterol, self-reported use of cholesterol-lowering medication, history of heart attack, and history of hypertension; results of multiple linear regression analysis. ^a^ Higher scores indicate poorer performance. Bolded p-values indicate statistically significant.

**Table 6 nutrients-16-00053-t006:** Sex-specific adjusted odds of poor performance * on cognitive function tests by egg consumption.

	Men		Women	
Cognitive Function Test	OR	(95% CI)	OR	(95% CI)
Mini-Mental Status Exam ≤ 24	0.978	(0.893–1.072)	0.936	(0.817–1.072)
Trails B ≥ 132	1.002	(0.941–1.066)	1.012	(0.956–1.071)
Category Fluency ≤ 12	0.959	(0.879–1.046)	0.967	(0.895–1.044)

* Categorical cutoffs indicative of poor performance Mini-Mental Status Exam ≤ 24; Trails B ≥ 132; Category fluency ≤ 12. Odds ratios were calculated with logistic regression; analyses adjusted for age, education, 1972–1974 smoking, cholesterol, use of cholesterol-lowering medication (in 1972–1974, history of heart attack, and history of high blood pressure. 95%CI = 95% confidence interval.

## Data Availability

Data analyzed for this study came from the publicly archived Rancho Bernardo dataset that can be found at https://knit.ucsd.edu/ranchobernardostudy/.

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
