# Peer review of "The Longitudinal Association of Egg Consumption with Cognitive Function in Older Men and Women: The Rancho Bernardo Study"

_nutrients, 2023, doi:10.3390/nu16010053_

Round 1

Reviewer 1 Report

Comments and Suggestions for Authors

The manuscript nutrients-2785690 is an article entitled The Longitudinal Association of Egg Consumption with Cognitive Function in Older Men and Women: The Rancho Bernardo Study by Donna Kritz-Silverstein examines the prospective association of egg consumption with multiple domains of cognitive function in older, community-dwelling men and women followed  for 16.3 years.

Participants were 617 men and 898 women from the Rancho Bernardo Cohort aged 60 and older who were surveyed about egg intake/week in 1972-74, and attended a 1988-91 research visit when cognitive function was assessed with 12 tests. Analyses showed that egg intake ranged from 0-24/week (means: men=4.2 ± 3.2, women=3.5 ± 2.7, p<0.0001). In men, covariate-adjusted regressions showed egg intake was associated with better performance on Buschke total (p=0.04), long-term (p=0.02) and short-term (p=0.05) recall. No significant associations were observed in women (p’s>0.05).

egg intake was positively associated with scores on Heaton copying(p<.04) and the Mini-Mental Status Exam (p<.02) in men, and category fluency (p<0.05) in women; it was not significantly associated with odds of poor performance on MMSE, Trails B, or category fluency in either sex.

These reassuring findings suggest there are no long-term detrimental effects of egg consumption on multiple cognitive function domains, and for men, there may be beneficial effects for verbal episodic memory.

The study is interesting, well described and well written

Methodology is appropriate.

Results are consistent.

Tables are informative

Conclusion are consistent with the results.

Major point: since eggs are lipid reach food, it would be of interest to know the evolution of the lipid metabolism.

A minor linguistic revision is recommended.

Minor point

Line 191: the table 1 is divided between two pages and it should be in one page

Line 216: table 2 should be in one page

Line 233: table 3 should be in one page

Line 251: table 4 should be in one page

Line 266: table 5 should be in one page

Comments on the Quality of English Language

minor linguistic revision

Author Response

We wish to thank the reviewer for their time, effort and laudatory comments on our manuscript describing it as interesting, well described, and well written.  Their comments and our responses are:

Major point: since eggs are lipid reach food, it would be of interest to know the evolution of the lipid metabolism.

Response: We are not clear what the reviewer meant by describing eggs as a “lipid reach” food and were unable to find that terminology in wither Pubmed or with a Google search. Perhaps the reviewer meant “lipid rich”? We have acknowledged that eggs are a lipid-rich food (lines 377-378).  However, while we appreciate the suggestion, we feel that a discussion of the evolution of lipid metabolism would be off focus of this paper. Most studies have reported that the cholesterol in eggs (which is dietary cholesterol) does not raise plasma cholesterol. We are trying to make the point that it is plausible that egg intake may be associated with better cognitive function  because eggs contain numerous nutrients and compounds that have been associated with better performance on cognitive function tests. 

A minor linguistic revision is recommended.

Response: Thank you for pointing this out.  We have read through the manuscript and made linguistic revisions as needed.  Wherever possible, these have been highlighted in yellow.

Minor point

Line 191: the table 1 is divided between two pages and it should be in one page

Line 216: table 2 should be in one page

Line 233: table 3 should be in one page

Line 251: table 4 should be in one page

Line 266: table 5 should be in one page

Response: We have revised the placement of the tables within the text so that each table appears on a single page rather than being divided into two pages.

Reviewer 2 Report

Comments and Suggestions for Authors

The manuscript written by Kritz-Silverstein and collaborators aims to examine the prospective association of egg consumption with cognitive function in older men and women resident in the community who were followed for 16.3 years. I congratulate the authors; a long investigation in time is never easy. I will try to make proper corrections so that the quality of the study can be improved.

- The introduction is well focused, but some hypotheses could be included.

- I believe that many more features of the participants and more noticeable differences between men and women can be added.

- The procedure section can be explained better, making a greater difference to the tests. Some explanatory figures would not be too much.

- It is worthwhile to include the procedural points in 2.2 and then continue with 2.3.

- Please include more current references; some of the most relevant are from 2011.

- Value to divide into paragraphs the last part of the discussion to give greater importance to practical applications and limitations.

Most of my contributions are advice, except for improving procedures. Once this is corrected the research is considered valid but there may be a lack of understanding at this point.

Author Response

Reviewer 2

We would like to thank Reviewer 2 for time, effort and helpful suggestions and comments on our manuscript.  Their comments and our responses are as follows:

- The introduction is well focused, but some hypotheses could be included.

Response: As suggested, we have included hypotheses. Please see the introduction, lines 72-73.

- I believe that many more features of the participants and more noticeable differences between men and women can be added.

Response: As suggested, several more features of the participants were added to Table 1, and men and women were compared on these features.  The baseline visit was fairly short and In general, we like to limit the presentation in a manuscript to those variables that are related to the analysis.  Nevertheless, as requested, we have added additional variables to Table 1. Please see Table 1, lines pertaining to self-reported heart attack, stroke, hypertension, and measured systolic and diastolic blood pressures.

- The procedure section can be explained better, making a greater difference to the tests. Some explanatory figures would not be too much.

Response: As requested, we have expanded the procedure section a bit in order to provide a better explanation of the cognitive function tests that were administered. However, we were unable to devise any explanatory figures that would add to the description rather than being redundant.

- It is worthwhile to include the procedural points in 2.2 and then continue with 2.3.

Response: Thank you very much for this suggestion. The procedure is now labeled as 2.2 and the statistics section is labeled as 2.3.

- Please include more current references; some of the most relevant are from 2011. 

Response: As requested, we have updated the references and added more recent ones. Please see highlighted references.

- Value to divide into paragraphs the last part of the discussion to give greater importance to practical applications and limitations.

Response: Thank you for this very useful suggestion.  We have now divided the paragraphs in the last part of the discussion into subsections.